# Mitochondria Can Cross Cell Boundaries: An Overview of the Biological Relevance, Pathophysiological Implications and Therapeutic Perspectives of Intercellular Mitochondrial Transfer

**DOI:** 10.3390/ijms22158312

**Published:** 2021-08-02

**Authors:** Daniela Valenti, Rosa Anna Vacca, Loredana Moro, Anna Atlante

**Affiliations:** Institute of Biomembranes, Bioenergetics and Molecular Biotechnologies (IBIOM)-CNR, Via G. Amendola122/O, 70126 Bari, Italy; r.vacca@ibiom.cnr.it (R.A.V.); l.moro@ibiom.cnr.it (L.M.)

**Keywords:** mitochondria, bioenergetics, oxidative phosphorylation, intercellular mitochondria trafficking, extracellular mitochondria, tunneling nanotubes, extracellular mitovesicles, ccf-mtDNA, neurodegenerative diseases, neurodevelopmental disorders, cancer, immune-metabolic regulation, mitochondrial transplantation

## Abstract

Mitochondria are complex intracellular organelles traditionally identified as the powerhouses of eukaryotic cells due to their central role in bioenergetic metabolism. In recent decades, the growing interest in mitochondria research has revealed that these multifunctional organelles are more than just the cell powerhouses, playing many other key roles as signaling platforms that regulate cell metabolism, proliferation, death and immunological response. As key regulators, mitochondria, when dysfunctional, are involved in the pathogenesis of a wide range of metabolic, neurodegenerative, immune and neoplastic disorders. Far more recently, mitochondria attracted renewed attention from the scientific community for their ability of intercellular translocation that can involve whole mitochondria, mitochondrial genome or other mitochondrial components. The intercellular transport of mitochondria, defined as horizontal mitochondrial transfer, can occur in mammalian cells both in vitro and in vivo, and in physiological and pathological conditions. Mitochondrial transfer can provide an exogenous mitochondrial source, replenishing dysfunctional mitochondria, thereby improving mitochondrial faults or, as in in the case of tumor cells, changing their functional skills and response to chemotherapy. In this review, we will provide an overview of the state of the art of the up-to-date knowledge on intercellular trafficking of mitochondria by discussing its biological relevance, mode and mechanisms underlying the process and its involvement in different pathophysiological contexts, highlighting its therapeutic potential for diseases with mitochondrial dysfunction primarily involved in their pathogenesis.

## 1. Introduction

Mitochondria are complex intracellular organelles long identified as the cellular power plants due to their vital role in oxidative energy metabolism. Acting as central metabolic hubs, mitochondria rapidly adapt to different environmental cues and metabolic alterations to meet the bioenergetics demands of the cell, recently defined as mitochondrial plasticity [1]. Owing to their highly plastic nature, mitochondria constitute a dynamic network of signaling organelles with multifunctional key roles in cell metabolism, proliferation and survival [2]. Notably, a variety of pre-clinical and clinical studies have demonstrated that metabolic and/or genetic mitochondrial alterations are involved in the pathogenesis of a large number of diseases, including cancer [3,4].

Mitochondria are highly interconnected entities ongoing fusion and fission events [5], with the capability to dynamically redesign their morphology as well as to move within the cell, thus supporting critical energy power needs [6]. Homeostasis of the mitochondrial pool is guaranteed by selective removal of mitochondria through mitophagy and by biogenetic mitochondrial supply, thus keeping an active and dynamic mitochondrial network within the cell [7]. In particular, in neurons, anterograde mitochondrial transfer from the cell body to the axonal extensions of the cell, or retrograde moving from axon to cell body can allow the removal of injured mitochondria or the restoration of healthy mitochondria. [8]. In invasive cancer cells, mitochondria relocate to the leading edge of the cells to provide fuel for their movement [9,10].

Emerging evidence is revealing that the dynamic nature of mitochondria can go beyond cell boundaries, allowing for their translocation between mammalian cells, radically challenging the hitherto known concepts of intracellular segregation of mitochondria and mitochondrial DNA (mtDNA) inheritance [11]. Their signaling role can extend to intercellular communication, showing that the mitochondrial genome and even entire mitochondria are indeed mobile and can mediate the transfer of information between cells. This mobile transfer task of mitochondria and mtDNA has been recently termed “momioma” to indicate all the “mobile functions of mitochondria and mitochondrial genome” [12]. Mitochondrial intercellular transfer promotes the integration of mitochondria into the endogenous mitochondrial network of recipient cells, contributing to changes in their bioenergetic status and other functional skills of receiver cells, not only in vitro but also in vivo [13]. Furthermore, the horizontal transfer of mitochondrial genes can lead to serious implications in the pathophysiology of mitochondrial dysfunction [14].

Although the physiological relevance of this phenomenon is still a matter of debate, several in vitro and in vivo studies have shown how the transfer of mitochondria among cells is able to recover mitochondrial respiratory defects in recipient cells, rescuing and regulating signaling, proliferation or resistance to chemotherapy, also functioning as means of tissue revitalization [15].

In this review, we will provide an overview of the up-to-date scientific data regarding the intercellular transfer of mitochondria, focusing on the cellular and molecular mechanisms mediating this process, discussing its biological relevance and involvement in different pathophysiological contexts—ranging from immuno-metabolic diseases, neurodegenerative/neurodevelopmental disorders and viral infections to cancer—and highlighting the potential therapeutic targeting of horizontal cell-to-cell transfer of mitochondria for treatment of diseases characterized by mitochondrial dysfunction.

## 2. Modes and Cellular Structures Mediating Intercellular Mitochondrial Transfer

The molecular mechanisms through which cells containing dysfunctional mitochondria acquire new mitochondria from other cells and the signaling pathways regulating this process remains yet poorly understood. The cells likely have developed mechanisms to trigger the transfer in response to the injury signals emanating from the recipient cell. However, the molecular signals initiating this type of crosstalk are yet unknown.

Several structures are involved in transcellular mitochondria transfer [16] (the main forms are depicted in Figure 1), among which tunneling nanotubes (TNTs) represent the main cellular system mediating intercellular mitochondrial translocation. Other modes of transfer have been identified, including membrane micro-vesicles, gap junctions, cell fusion or mitochondrial expulsion. Mitochondrial transfer through these different structures can lead to different functional outcomes for the recipient cells, i.e., functional mitochondrial acquisition, immune activation or trans-mitophagy [14]. Therefore, a clear understanding of the mechanisms that mediate mitochondrial transfer will shed light on how this process is regulated and can be exploited for therapeutic purposes.

TNTs are nanotubular structures generated through formation of cell membrane protrusions that attach to the target cell. The membrane of each cell extends to fuse together, thus forming a firmly connected bridge suspended in the extracellular space [17]. TNTs allow unidirectional and bidirectional transport of a variety of components, including small molecules, proteins, organelles, and even viral particles [18,19,20].

The discovery of TNTs in 2004 appeared as an innovative cell-to-cell communication mechanism, demonstrating the ability of mammalian cells to exchange organelles with other cells [21]. TNTs are dynamic structures forming de novo in a few minutes and with a half-life ranging from a few minutes to several hours [21,22]. TNTs have a skeleton composed mainly of F-actin and transport proteins that assist the active transport of cargoes and mitochondria along these structures [23]. TNTs act as channels for mitochondrial transfer between various cells. In particular, under conditions of oxidative stress, the intracellular expression of p53 is upregulated and the protein kinase B-phosphoinositide 3-kinase-mechanistic target of rapamycin (AKT-PI3K-mTOR) signaling pathway is activated, leading to the formation of TNTs between stressed and unstressed cells mediating the transcellular transport of several organelles, including ER, Golgi, endosomes and mitochondria [24]. In the bone marrow microenvironment of multiple myeloma, TNT-mediated transcellular transfer of mitochondria from neighboring nonmalignant bone marrow stromal cells to multiple myeloma cells supports oxidative phosphorylation of multiple myeloma cells and is dependent on the expression of CD38 [25]. TNTs also form between tumor cells and have been implicated in their survival and drug resistance [26,27].

In neurons, the molecular motor myosin X is also required for TNT formation [28]. A study picked out the participation of an extracellular protein, S100A4, and its receptor RAGE (Receptor for Advanced Glycation End Product) in managing TNT growth direction in the nervous tissue [29]. Under oxidative stress, hippocampal neurons and astrocytes initiated the formation of TNTs after p53-mediated activation of caspase-3. S100A4 cleavage by activated caspase-3 formed a gradient of low levels of S100A4 in initiating damaged cells towards a higher concentration in receiving cells, i.e., astrocytes [29]. The results of this study disclose that injured neurons transfer cellular contents to astrocytes, scattering the danger signals and inducing mitochondrial transfer. TNT-mediated mitochondrial transfer can act as a survival way for cells under stress, for example, by saving damaged ischemic cells [30], by protecting the alveolar epithelium from injury [31] and repairing the tissues [32,33,34].

### 2.1. Mitochondrial Transfer via Extracellular Vesicles

A heterogeneous population of vesicles (ranging from 40 to 1000 nm) are generally released from the intracellular to the extracellular environment. These vesicles are denoted as extracellular vesicles (EVs). They include exosomes (30–100 nm in diameter), micro-vesicles (MVs) (100 nm to 1 μm in diameter) and apoptotic bodies (1–2 μm) depending on their source and molecular structure [35]. Apoptotic bodies are the least studied due to their rapid elimination by phagocytic cells [36]. Exosomes and micro-vesicles were originally defined as vesicles to eliminate ancient proteins in immune cells and reticulocytes [36], but, more recently, it has been shown that they are released from nearly all cell types and considered responsible for communication between cells in a variety of pathophysiological events [37].

EVs containing lipids, proteins, RNA and mitochondria represent a competent mode to transport functional loads from one cell to another [38]. This discovery has introduced a new mode of interactions to an already multifaceted communication network and a new signal transmission mechanism [39].

Although the mechanisms by which mitochondrial proteins or mtDNA are loaded into EVs are still unknown, surely mitochondrial components have been detected in EVs. If smaller EVs such as exosomes can mainly carry small RNAs [40], but genomic DNA and mtDNA have also been detected [41,42], larger EVs such as MVs can enclose even entire mitochondria [43]. MVs carrying mitochondria can be secreted by different cell types, as observed in astrocytes, neurons and mesenchymal stem cells [41,44,45,46,47] and transferred into epithelial cells, immune cells, astrocytes and neurons [11]. The mitochondrial transfer does not always keep damaged cells, but also works to recover organelles into other cells by a transcellular degradation process [44,45]. Furthermore, Morrison et al. showed that mesenchymal stromal cells (MSCs) modulate macrophages, improving their respiration and phagocytic activity in clinically relevant lung injury models by EVs containing mitochondria [47]. Importantly, these exosome-mediated transferred mitochondria co-localize with the mitochondrial network and produce reactive oxygen species (ROS) within the recipient T cells [46].

### 2.2. Transfer via Gap Junction Channels

Gap junction channel (GJC) is a structure joining the cytoplasm of two separate cells. GJCs allow the passive transport of nutrients, metabolites, second messengers, cations, anions and also mitochondria [31,48,49]. Cx43 is a connexin that participates in the transcellular transfer of mitochondria. In a model of lipopolysaccharide (LPS)-induced acute lung injury, bone marrow stromal cells (BMSCs) transport mitochondria to the damaged alveolar epithelium, a process based on Ca^2+^ exchange between the two cells through CX43-GJCs [31]. Recently, it was shown that the transfer of mitochondria from BMSCs to hematopoietic stem cells (HSCs) is a timely physiological event in the mammalian response to acute bacterial infection [47]. Mechanistically, oxidative stress was found to regulate the opening of connexin channels in a PI3K-mediated system, the activation of which allowed the transfer of mitochondria from BMSCs to HSCs [46]. Cx43-GJCs participate in the intercellular exchange of ROS [50] and a direct mitochondrial transfer via Cx43 GJC has been hypothesized [16], which could represent a mechanism for TNT formation and intercellular transport of mitochondria.

### 2.3. Mitochondrial Transfer via Other Routes: Mitochondrial Extrusion and Cell Fusion

In addition to TNTs, EVs and Cx43-GJCs, representing the main routes that mediate transcellular transfer of mitochondria, other potential mechanisms for mitochondrial transfer from one cell to another are the mitochondrial extrusion and the cytoplasmic fusion.

The mitochondrial extrusion allows the release of mitochondria or mitochondrial components from cells under specific conditions during which mitochondria become inadequate to remain in cells. Retaining damaged mitochondria can produce large quantities of ROS [50] and under such circumstances, cells tend to dispose mitochondria extrusion into the intercellular space [51]. Naked mitochondria or mitochondrial components can also be extruded and internalized without carrier, through processes of exocytosis and endocytosis [15]. Mitochondrial extrusion takes place not only in vitro but also in vivo. Few studies have shown the release of naked or encapsulated mitochondria into the extracellular environment. Nakajima et al. confirmed in a mouse model the release of naked mitochondria into the intercellular space after an injection of anti-Fas antibodies [52]. In response to this treatment, the cytoplasmic vacuoles englobed the fragmented mitochondria and extruded them from the apoptotic hepatocytes. Similarly, activated platelets released respiration-competent mitochondria, as either free organelles or mitochondria engulfed within microparticles [52].

Cytoplasmic fusion is a common phenomenon in which the membrane of two or more cells fuse together, sharing the organelles, when injury and inflammation may trigger this process [53]. In particular, cell fusion regulates the potential of stem cells, playing a significant role in regeneration and oncogenesis [54]. Mitochondrial transfer takes part in this process [16].

## 3. Intercellular Mitochondrial Transfer in Different Pathophysiological Conditions

### 3.1. Transcellular Mitochondrial Transfer in Nervous Cells 

In the central nervous system (CNS), mitochondrial transfer constitutes an important form of intercellular crosstalk contributing to the homeostasis and neuroprotection of CNS [45]. The transfer of mitochondria among nervous cells participates in the cellular and tissue defense against CNS damage, performing a relevant function in recovery after injury [45]. In the CNS, astrocytes perform a wide range of functions, including the regulation of neurodevelopment, neurotransmission and metabolism (reviewed in [14]). Emerging studies show that the neuroprotective functions of astrocytes may comprise the transfer of mitochondria from these cells to damaged neurons [45,55,56]. Under physiological conditions, astrocytes defend neurons, counteracting oxidative stress and excitotoxicity and ensuring neurotrophic support [57,58]. In the context of injury, astrocytes can transfer healthy mitochondria to axons [45,59]. Recently, a study of Cheng et al. in co-culture systems of human-induced pluripotent stem cells (iPSC) showed that iPSC-derived astrocytes can transfer mitochondria to neurons, recovering dopaminergic neuronal toxicity [60]. The neuroprotective function of astrocyte-derived mitochondria transfer to neurons has been highlighted in other studies of neurotoxicity; for instance, English et al. demonstrated, in their in vitro work on a co-culture of cisplatin-treated neurons with astrocytes, that neurons that had received mitochondria from astrocytes showed improved neuronal survival, reinstated neuronal mitochondrial membrane potential and normalized neuronal calcium dynamics, underlining the relevance of transcellular mitochondrial transfer [61]. Notably, under stress conditions, astrocytes may interrupt their protective function and release danger signaling factors, such as inflammatory cytokines, thus damaging neurons (reviewed in [14]).

In broader terms, there is growing evidence revealing that like in the whole CNS, transcellular transfer of mitochondria can have significant relevance and multifunctional roles in various physio-pathological contexts:The transfer of mitochondria from neurons to astrocytes can activate a process known as trans-mitophagy, allowing cells to degrade dysfunctional mitochondria [45];The transfer of astrocytic mitochondria to injured neurons can lead neuroprotective benefits [45,55,56];The transfer of active mitochondria from endothelial progenitor cells to brain endothelial cells can enhance cell viability and improve their function of defensive barrier [62,63];The transfer of mitochondria from hematopoietic stem and progenitor cells to neurons can improve their mitochondrial functional efficiency [14].

### 3.2. Mitochondrial Transfer in Dysfunctional Mitochondria-Related Neurodegenerative Disorders: Therapeutic Use of Exogenous Mitochondria for Alzheimer’s and Parkinson’s Diseases

Pathogenesis of neurodegenerative disorders, such as Alzheimer’s and Parkinson’s diseases, involves dysfunction of mitochondria [64,65].

Alzheimer’s disease (AD) is the most common neurodegenerative disorder, characterized by a progressive failure in cognitive function due to progressive loss of neurons in forebrain and other brain areas [66]. Mitochondrial dysfunction has been established as an early and prominent feature of the disease [67]. The multiform, and even opposed, modes involving mitochondrial dysfunction in AD pathophysiology and their complex regulation make the aim of targeting mitochondrial deficits very difficult. Nitzan and coworker’s experimental strategy aimed at overcoming this limitation by using active and functional mitochondria, thereby allowing mitochondria to act as whole organelles rather than targeting only one of their dysfunctional tasks. The results of their analysis suggest that transfer of functionally active mitochondria, aimed at efficiently mimicking mitochondrial function, is beneficial to treat AD deficits, correcting cognitive deficits, brain pathology and mitochondrial defects in an AD mouse model [68]. In this recent in vivo study, the effect of transferring active intact mitochondria was investigated, by treating AD-mice (amyloid, intracerebroventricularly injected) intravenously (IV) with fresh human isolated mitochondria. Fourteen days after mitochondrial transplantation, AD-mice treated with exogenous mitochondria showed significantly improved cognitive performances almost comparable to those of untreated control mice [68]. A significant recovery in neuronal loss and reduced gliosis were also detected in the hippocampus of treated mice respect to untreated AD-mice. Increased citrate synthase and cytochrome *c* oxidase activities were measured in mitochondria-treated AD-mice, reaching activity values close to untreated control mice [68]. Increased mitochondrial activity was also detected in the liver of mitochondria-treated mice. No toxicity associated with the treatment was detected. Therefore, mitochondrial transfer could offer a novel therapeutic approach for AD treatment.

Parkinson’s disease (PD) is the second most common neurodegenerative disorder, characterized by the selective loss of dopaminergic neurons of the substantia nigra pars compacta (SNc) with motor and nonmotor symptoms [69]. The main histopathological marker of PD is the presence in neurons of α-synuclein (α-syn) protein aggregates forming in inclusion bodies, indicated as Lewy bodies [70]. α-Syn is principally expressed pre-synaptically and there is evidence of the existence of α-syn transfer from neurons to neuronal and non-neuronal cells in vitro, indicating that α-syn pathology propagates between anatomically adjacent brain regions by an intercellular transfer mode [71]. Mitochondrial dysfunction is widely recognized as a common central pathway involved in the pathogenetic processes of sporadic and genetic PD (reviewed in [72]). Dysfunctional mitochondria are constant presences in PD [73]; moreover, α-syn can be located at mitochondrial membranes and its aggregation can be related to mitochondrial dysfunction in PD [74]. Increased ROS levels resulting from reduced efficiency in the electron transport chain activity are involved in the formation of α-syn aggregates and neuronal loss [75].

Increasing evidence suggests that astrocytes have a relevant part in the progression of PD (reviewed in [72]). A recent work presented clear evidence that transneuronal mitophagy occurs in vivo in PD models [76]. In PD models, astrocytes are primarily responsible for clearance of damaged mitochondria—a functional role of considerable relevance in the context of PD associated to mutations of Parkin and PINK1 [72]. Notably, PINK1 activity was recently predominantly found in astrocytes while almost absent in neurons [77]. Exogenous supplementation of mitochondria to damaged regions may be a potential and innovative therapeutic strategy for the treatment of PD, as shown in the in vivo work of Chang et al., demonstrating that injection of mitochondria into medical forebrain bundle (MFB) of 6-hydroxydopamine-unilaterally infused PD rats enhanced the survival of dopaminergic neurons and improved mitochondrial functions by recovering normal levels of mitochondrial complex I-IV and reducing mitochondrial oxidative stress in vivo [78].

Remarkably, the translational application of mitochondrial transfer should be further evaluated and its therapeutic potential exploited for the treatment of neurodegenerative diseases, such as AD and PD.

### 3.3. Mitochondrial Transfer in Neurodevelopmental Diseases: Extracellular Mitochondrial Release Reflecting Mitochondrial Dysfunction in Down Syndrome and Fragile X Syndrome

Mitochondrial dysfunction is a critical player contributing to the pathogenesis of several neurodevelopmental diseases, including Down syndrome (DS), the most common genetic defect leading to intellectual disability and caused by the trisomy of human chromosome 21. DS is characterized by neuropathological changes occurring already in fetal and neonatal life, leading to alterations in brain development [79].

Defective mitochondrial bioenergetics negatively compromise neuronal development and represent an early event in developing the neurobiological alterations characterizing the syndrome [80,81,82]. Although DS is a multi-genic disorder and many pathways are affected, oxidative phosphorylation (OXPHOS) dysfunction was found ubiquitously present in any tissue or cell type, regardless of age, including fetal one, so that DS is now regarded as an OXPHOS disorder [83].

In a very recent study, a new population of extracellular vesicles containing mitochondrial proteins, named “mitovesicles”, were identified and found altered in Down syndrome [84]. D’Acunzo et al. have shown that brain-derived mitochondria contain a specific subset of mitochondrial components and that their levels and cargo are aberrant in DS [84]. Comparative analysis of EVs derived from brains of Ts2 mouse model of DS and obtained from post-mortem human brains of individuals with DS showed higher numbers of mitovesicles with altered composition in the DS brain parenchyma, in both murine and human post-mortem brains [84] (Figure 2A). These data indicate that mitochondrial damage directly affects mitochondrial biology, either by activating the release of these vesicles or by regulating the mitochondrial cargo in the single extracellular vesicle.

Taken together, these data show that mitochondrial levels and composition of mitovesicles mirror mitochondrial alterations within the cell of origin and could be used as biomarkers for assessing mitochondrial brain dysfunctions in neurological disorders.

Mitochondrial dysfunction contributes to the pathogenesis of another neurodevelopmental disease, the Fragile X syndrome (FXS). FXS is an inherited disorder characterized by mental retardation, caused by silencing of the *fmr1* gene, encoding the Fragile X mental retardation protein (FMRP) [86], an RNA-binding protein expressed mainly in neurons and astrocytes of the brain and associated with approximately 4% of transcripts, including those for mitochondrial proteins [87]. Neuronal development in *Fmr1* knock-out (KO) mice exhibited impaired dendritic maturation, altered expression of mitochondrial genes, fragmented mitochondria, impaired mitochondrial function and increased oxidative stress [88].

D’Antoni et al. provided the first evidence of a compromised and inefficient mitochondrial bioenergetics in the brain cortex of *Fmr1* KO mice, a model of FXS, supporting the idea that mitochondrial dysfunctions may play a critical role in pathogenesis of the syndrome [89].

In a recent study, the ability of EVs to transfer mitochondrial components and their role in mitochondrial dysfunction was assessed in astrocytes and brain cortices from *Fmr1* KO mice FLX model [85]. The mitochondrial protein levels of the transcription factor NRF-1 (nuclear respirator factor 1), the subunits ATP5A and ATPB of ATP synthase and the mitochondrial membrane protein VDAC1 in EVs were found drastically reduced in cerebral cortex and astrocyte samples from *Fmr1* KO mice compared to euploid mice. These reductions are related to a reduction in mitochondrial biogenesis in the *Fmr1* KO brain, associated with decreased mitochondrial membrane potential in *Fmr1* KO astrocytes. Mitochondrial components were found reduced in both EVs derived from cerebral cortices and those secreted from astrocytes of *Fmr1* KO mice (Figure 2B). The depletion of mitochondrial proteins contributes to mitochondrial dysfunction in astrocytes [85]. This study indicates that mitochondrial dysfunction in astrocytes is related to the pathogenesis of FXS and can be monitored by depletion of EV mitochondrial components.

These findings may improve the ability to diagnose neurodevelopmental diseases associated with mitochondrial dysfunction. However, this kind of study is nascent and further investigations are needed to define the exact mechanisms responsible for the observed decrease in mitochondrial proteins, which could mirror a deficit in intracellular transferring from mitochondria to EVs or a compromised EVs formation.

Furthermore, to better understand the complex physiological functions of astrocyte-derived EVs, it will be crucial to determine which components of EVs critically impact the progression of FXS disease and its regulatory mechanism.

### 3.4. Mitochondrial Transfer in Tumorigenesis and Chemotherapy Resistance

Cancer cells take advantage of intercellular mitochondrial transfer to support their metabolic needs, survival and chemoresistance.

As Warburg hypothesized more than 80 years ago, tumor cells are typically inclined to upregulate glycolysis [90]. This behavior might seem counterintuitive, as glycolysis is energetically inefficient compared to OXPHOS and acidifies the extracellular microenvironment by increasing lactate production and extrusion. However, the metabolic shift towards glycolysis would efficiently provide the intermediate building blocks to support the increased biosynthetic needs of cancer cells. Even though most cancer cells have a preference for glycolysis, increasing evidence indicates that certain solid tumors and many hematological malignancies display normal or even increased OXPHOS and mitochondrial metabolism [91,92]. Indeed, recent studies have pointed out the relevance of mitochondrial-dependent metabolic reprogramming in enhancing cancer cell proliferation and survival, and in developing chemoresistance in many cancer types [93,94].

Cancer cell mitochondria play a pivotal role in the mutual interaction of neoplastic cells with the tumor microenvironment [95]. As pointed out by recent findings, tumors comprise not only malignant cells, rather, they are a complex system of tumor and non-tumor cells that create a symbiotic relationship within the tumor microenvironment to foster survival and chemoresistance [96].

Cancer cells are able to extrude the whole mitochondria or some of their constituents, including mtDNA, cytochrome *c* and formylated peptides, in the tumor microenvironment, [97], which, in turn, function as Damage-Associated Molecular Patterns (DAMPs) that activate the innate immune system [98,99]. Although DAMPs activate the host’s defense system, they also promote pathological pro-inflammatory and immunosuppressive responses that can drive cancer cell proliferation and invasion [100].

Mitochondria-dependent intercellular communication between cancer and non-cancer cells through cell–cell contacts [101] and secretion of soluble factors and extracellular vesicles [102] represents a key mechanism for cancer cells to escape immune surveillance and develop chemoresistance [93,94]. Notably, horizontal transfer of mitochondria between cancer and non-cancer cells may play a central role in driving malignant progression [25,103,104,105].

The first demonstration of horizontal mitochondrial transfer was reported by Spees and coworkers in 2006: A549 cells devoid of mtDNA (A549 ρ° cells) co-cultured with human mesenchymal stromal cells (MSCs) or skin fibroblasts were able to restore mtDNA content and a functional mitochondrial pool [101]. In their studies, Berridge and Tan demonstrated that metastatic murine melanoma (B16) and breast cancer (4T1) ρ° cells were able to recover functional mitochondria from the tumor microenvironment, thus restoring OXPHOS and tumorigenicity to the levels of parental cells [106,107]. It was later demonstrated the occurrence of mitochondria transfer from MSCs to ρ° cells to restore mtDNA, OXPHOS and ability to form tumors [104].

In a typical horizontal mitochondrial transfer, recipient cells are characterized by elevated OXPHOS needs and/or severely compromised mitochondrial functionality [25,108,109], while donor cells are proficient in mitochondrial functionality and appropriately activated [105,110]. So far, only a few molecular mediators involved in horizontal mitochondrial transfer have been identified. Among these, metalloproteinase-1 (MMP-1), nestin and proinflammatory cytokines are crucial mediators that stimulate donor cells to transfer mitochondria [105]. In addition to these factors, PGC1α (a master regulator of mitochondrial biogenesis) is implicated in mitochondrial transfer from donor MSCs to recipient leukemic cells [111]. Furthermore, activation of the donor cells is associated with an increase in ROS levels mediated by the recipient cells [105,110,112], suggesting that ROS represent one of the mediators of the directional mitochondrial transfer.

In solid tumors, highly glycolytic cancer-associated fibroblasts (CAFs), which are implicated in the metabolic reprogramming of cancer cells [113], tend to donate their mitochondria to prostate cancer cells nearby, thus stimulating cancer cells’ OXPHOS [114]. These findings suggest that mitochondria transfer from CAFs is another pathway allowing the metabolic plasticity of cancer cells, which could support tumor progression.

A debate still exists about the significance of restoring mitochondrial respiration in ρ° tumor cells to drive their tumorigenic potential. A recent study indicates that the primary reason is not “energy” or “more ATP” needed; instead, mitochondrial respiration would provide dihydroorotate dehydrogenase (DHODH), an essential intermediate for the de novo synthesis of pyrimidines [26]. Consistently with this hypothesis, deletion of DHODH in tumor cells with fully functional OXPHOS suppressed tumor development, while suppression of mitochondrial ATP synthase produced minimal effects [26], thus implicating DHODH as a potential therapeutic target for OXPHOS-dependent cancers.

Chemoresistance to cancer therapy is still a critical issue to evaluate in order to ensure the efficacy of therapeutic treatment. Many studies have hypothesized the involvement of several potential responsible mechanisms that include intrinsic and extrinsic processes, the latter being greatly affected by intra-tumor heterogeneity [115]. In particular, a significant factor leading to intra-tumor heterogeneity is the presence in the tumor microenvironment of many non-malignant cells recruited to the tumor site, such as CAFs, MSCs and immune cells [116,117]. Intercellular communications and interactions between malignant and non-malignant cells are pivotal in tumor heterogeneity and chemoresistance [115]. For instance, MSCs isolated from bone marrow specimens of patients with acute lymphoblastic leukemia (ALL) converted to an activated cancer-associated fibroblast phenotype when treated with the chemotherapeutic drugs cytarabine and daunorubicin, and prevented therapy-induced apoptosis of ALL cells by transferring functional mitochondria through TNTs [118].

The intercellular transfer of mitochondria represents an intriguing mechanism, still partially understood, whose targeting may provide new avenues in cancer therapy. Evidence that mitochondrial transfer may occur similarly in solid and hematological tumor cells further increases the importance of this process. Furthermore, the involvement of mitochondrial transfer during cancer progression and development of chemoresistance may explain the yet unclear mechanisms of action of certain anti-cancer drugs.

### 3.5. Mitochondrial Transfer in Immune-Metabolic Regulation

Recent evidence suggests a role of intercellular mitochondria transfer in the regulation of the immune system [119]. Court and coworkers reported that the incorporation of exogenous mitochondria promotes the programming of regulatory T cells in the steady state, suggesting that the transfer of mitochondria could have anti-inflammatory properties [119].

#### 3.5.1. Mitochondrial Transfer from MSCs to Macrophages and T Cells

During the tissue repair process, macrophages play a vital role in eliminating inflammatory products through phagocytosis. Macrophages are cells of the immune system that play a fundamental role in the inflammatory response, characterized by a spectrum of different polarization and activation states, the extremes of which are represented by the “classically activated” and pro-inflammatory macrophages M1 and the “alternately activated” and anti-inflammatory M2 macrophages. MSCs can improve the anti-inflammatory capability of macrophages by inducing their differentiation in M2 phenotype [120].

Several in vitro and in vivo studies have shown that mitochondria transported from MSCs to macrophages can induce the selective differentiation of macrophages towards the anti-inflammatory M2 phenotype and contribute to the antimicrobial effect of MSCs [47,121]. In acute respiratory environmental distress syndrome, OXPHOS activity and phagocytosis of macrophages were stimulated after they acquired functional mitochondria from MSCs [121,122], and it has been hypothesized that stimulated OXPHOS was responsible for the conversion of the M2 phenotype of macrophages [47]. In turn, inhibition of intercellular mitochondrial transfer by affecting the mitochondria of MSCs [47] or by blocking the transfer pathway [121,122] prevents phagocytosis and bioenergetics of macrophages.

In addition to macrophages, pathogenic T helper 17 (Th17) cells also can take up mitochondria from bone marrow-derived MSCs in a co-culture system, which increased oxygen consumption and decreased IL-17 production by Th17 cells [123]. Furthermore, poor mitochondrial transfer to Th17 cells from synovial stromal stem cells was observed in patients with rheumatoid arthritis compared to bone marrow MSCs isolated from healthy donors [123].

In a recent study, the effect of intercellular mitochondrial transfer and direct transplantation of MSC-derived mitochondria to peripheral blood lymphoid mononuclear cells was assessed [119]. Mitochondria isolated from transplanted MSCs have been shown to be acquired inside cells with increased expression of mRNAs associated with activation of T lymphocytes and differentiation of regulatory T cells (Treg) (FOXP3, IL2RA, CTLA4 and TGFβ1), leading to an increase in the number of Treg cells and a consequent immunosuppressive effect [119]. Therefore, intercellular mitochondrial transfer could be a novel target for MSCs that might be used to facilitate immunoreactions and treat immune diseases.

#### 3.5.2. Mitochondrial Transfer from Adipocytes to Macrophages as Immune-Metabolic Crosstalk Regulating Metabolic Homeostasis: Impairment in Obesity

Metabolic crosstalk between adipocytes and immune cells are essential to guarantee tissue homeostasis and, when dysregulated, can induce pathological inflammation that triggers obesity and obesity-associated metabolic dysfunction [124,125,126,127].

The transfer of intercellular mitochondria takes place in vivo in the white adipose tissue (WAT) by means of a mitochondrial transfer axis from adipocytes to macrophages found dysregulated in obesity [128]. Brestoff and coworkers also showed that the efficiency of this process is reduced in nutritional conditions of diet-induced obesity due to the reduced intake of mitochondria by the macrophages contained in the WAT [128] (Figure 3).

Macrophages that have acquired mitochondria from adipocytes overproduce mitochondrial ROS and show signals of hypoxia and de-enrichment of nuclear encoded genes involved in mitochondrial homeostasis and maintenance of the electron transport chain [128]. In comparison, the occurrence of mitochondrial transfer from adipocytes to macrophages is markedly reduced in obesity, a pro-inflammatory state in which macrophages are exposed to factors that induce a type 1 immune response (M1) [125,129,130,131,132,133].

The directional transfer of mitochondria to macrophage frequently occurs in healthy WAT when macrophages are biased towards an M2 activation state driven by type 2 cytokines, such as IL-4 and IL-13, which are produced by innate group 2 lymphoid cells, eosinophils and other cell types present in healthy WAT [125,127]. This suggests that M1-like polarization inhibits entry of mitochondria. These remarks imply that a reduction in mitochondrial transfer is a feature of metabolic diseases such as obesity, at least in the WAT.

The mechanisms involved in mitochondria transfer into cells are not yet well understood. The uptake of mitochondria is inhibited by cytochalasins that impair the rearrangement of the actin cytoskeleton [125,134]. By performing genomic screening, Brestoff and coworkers identified 23 candidate genes that could contribute to mitochondrial uptake. Among these, 13 genes are involved in the heparane sulfate (HS) biosynthesis pathway [128].

Intercellular mitochondrial transfer occurs, at least in part, through an HS-dependent mechanism both in vitro and in vivo, and deletions of genes involved in HS synthesis compromises intercellular mitochondrial transfer to macrophages and alters energy homeostasis, thus suggesting the possibility of a functional correlation. Indeed, patients and mice with heterozygous mutations associated to loss-of-function in *Ext1* (exostosin 1) encoding the copolymerase involved in HS biosynthesis [135] manifest impaired lipid and glucose homeostasis [136].

The transfer of intercellular mitochondria to macrophages could participate in the maintenance of energy homeostasis in mice. These findings further reinforce the possibility that the obesity-associated reduced transfer of mitochondria from adipocytes to macrophages may have a part in weight gain. These results suggest a new pattern of immune-metabolic cross-talk between cells, in which some cells, such as adipocytes, translocate their mitochondria to macrophages to control systemic metabolic homeostasis.

Targeting the intercellular mitochondrial transfer could become a new therapeutic strategy for the treatment of metabolic diseases.

## 4. Extracellular Mitochondria: Active Players in Health and Disease

As described so far, mitochondria can be present in the extracellular space in intact and free form (freeMitos), or enclosed by a membrane, as inside platelets or vesicles, or as cell free circulating mtDNA (ccf-mtDNA), with different possible roles ranging from producing recovery effects to acting as a danger signal when interacting with other cells.

The interactions of these extracellular mitochondria with other cells open a new area of study in which mitochondria goes beyond their role as cell powerhouses, becoming, per se, signaling organelles [11,137]. Knowing the role of extracellular mitochondria and their different forms could favor the development of new therapeutic approaches to improve health as well as identify new biomarkers of diseases.

### 4.1. Extracellular Circulating Cell-Free Mitochondria in Blood

Several indications show the presence of circulating cell-free circulating mitochondria in the blood released from numerous cell types under conditions of stress, injury or disease [137]. Recently, Stephens and coworkers showed cell-free mitochondria release under non-pathologic contexts and their detection as circulating mitochondria in murine and human blood [138]. The authors used flow cytometry to detect circulating mitochondria in platelet-depleted plasma in healthy mice and humans, demonstrating that circulating cell-free mitochondria have an active transmembrane potential and were able to enter ρ° cells. A proteomics study recognized mitochondria-specific and EV-associated proteins in sorted circulating cell-free human mitochondria [138]. These findings fit with a recent publication showing the presence of cell-free mitochondria displaying normal O_2_ consumption in the blood of healthy human subjects [139].

The authors provided an evaluation of 200,000 up to 3,700,000 respiratory-competent mitochondria per ml of extracted plasma, and suggested that circulating cell-free respiratory-competent mitochondria could represent a novel class of signaling organelles involved in regulatory activities and intercellular communication. Although the evidence for the presence of cell-free mitochondria in human blood is convincing, the conclusion that these mitochondria are energy competent or functional for respiration has been more recently refuted [140]. In its report, Stein evaluated the functionality of cell-free mitochondria in human blood using high-resolution respirometry and mitochondria extracted from platelets from the same blood samples as positive controls. Although cell-free mitochondria were present in human plasma, there was no evidence proving that their mitochondrial electron transport (ETS) system was functional, as shown by evaluating parameters such as the respiratory rate, which was found not significantly different from 0, as well as no significant response to ADP and a lack of sensibility to uncoupler and/ or OXPHOS inhibitors. However, the activity of the complex IV in vitro was detectable and even slightly higher than the levels found in mitochondria extracted from platelets, suggesting that cell-free mitochondria in human blood are likely to retain only a non-functional part of the ETS. Although there are doubts that they are functionally capable of OXPHOS activity, circulating mitochondria may have significant physiological roles, which need to be explained.

Characterization of these circulating mitochondria is important to define a baseline in healthy individuals that will allow for comparison in pathological states.

Defining origins and functions of extracellular mitochondria will be crucial for understanding their overall impact on health and disease.

### 4.2. Extracellular Cell-Free Circulating Mitochondrial DNA as an Alarm Signal

Mitochondria are the only intracellular organelles, outside the nucleus, that house their own genome—the mtDNA. In mammals, mtDNA is an intron-less circular double-stranded molecule maternally inherited and present in multiple copies per cell [141]. The mitochondrial genome consists of 37 genes—13 encoding protein subunits of OXPHOS machinery producing ATP, 2 rRNAs (12S and 16S) and a complete set of 22 tRNAs for mitochondrial protein synthesis [142]. The absence of efficient DNA repair systems and protective histones present in the nuclear DNA makes mtDNA more susceptible to mutations [143]. Physical stressors, such as trauma, infection or strenuous exercise can promote mtDNA extrusion first from mitochondria to the cytosol, and then from the cytosol to the microenvironment until it reaches the circulation as circulating cell-free mtDNA (ccf-mtDNA) [98,144,145,146]. Owing to the bacterial origin of mtDNA, ccf-mtDNA is still recognized as foreign in the body (immunogenic) and, consequently, it triggers an inflammatory response [98,147]. When ccf-mtDNA is released by lymphocytes, it triggers immune activation [148], suggesting the existence of cell type-dependent mechanisms to regulate mtDNA release. mtDNA can be extruded from mitochondria that accumulate damage, acting as a trigger for the immune cells to activate elimination of damaged cells and either stimulate pro-inflammatory signals [149,150,151,152] or allow recombination and mixture with the genome of the recipient cells upon transportation in EVs [11,153,154].

MtDNA-induced inflammation following severe trauma in patients with multiple organ failure causes the presence of elevated levels of ccf-mtDNA [155,156]. In addition, high ccf-mtDNA levels have been also detected in patients with diabetes, cancer and myocardial infarction [152], making ccf-mtDNA levels a potential prognostic biomarker in these pathological conditions.

ccf-mtDNA is commonly considered a danger signal, acting like Damage Associated Molecular Pattern (DAMP) molecules [16,97]. The signals generated by mitochondrial-derived DAMPs may be chemotactic, phagocytic, immune-stimulatory or regenerative. For example, mtDNA DAMPs scan stimulate neutrophils, thus activating additional immune cells that cause an inflammatory response by releasing pro-inflammatory cytokines that can propagate the damage to distant tissues/organs [157]. Further understanding of ccf-mtDNA and mtDNA DAMPs in the etiology of different diseases could advance our knowledge on the inflammatory response and may eventually provide novel molecular tools for the development of anti-inflammatory therapies.

### 4.3. Mitochondria as Critical Mediators and Potential Therapeutic Targets in Viral Infection: Focus on SARS CoV-2 Coronavirus

A large body of studies has shown that a variety of viral infections can alter the mitochondrial function and cellular innate immune response to maintain intracellular survival (recently reviewed in [158]). Viruses can manipulate cell energy metabolism, reprogram metabolic pathways and use metabolites to maintain viral niches in the cells [158].

Interestingly, TNTs have been widely implicated in promoting viral infection spreading in different steps of their cycle of infection [159]. Many viruses, including the flu virus, the human immunodeficiency virus (HIV) and the herpes simplex virus, use TNTs to transfer their genetic material within the target cells, followed by manipulation of the mitochondrial functionality [159]. In this way, viruses can evade host immunity and bypass pharmaceutical targeting aimed at preventing their entry into the cells through plasma membrane receptors [20,160,161]. For instance, human macrophages present in lymph nodes derived from HIV-infected individuals displaying HIV reactivation exhibited TNT-like structures [162]. As discussed in a recent review [162], TNT-mediated viral spread may accelerate and exacerbate the viral disease by targeting multiple organs and tissues, as described for severe cases of SARS-CoV-2-mediated coronavirus disease 2019 (COVID-19).

The complex pathological condition resulting from the pandemic SARS CoV-2 infection presents a broad spectrum of clinical manifestations leading to development of acute respiratory syndrome, accompanied by cytokine storm and multiorgan failure in severe cases [163]. Although the mechanisms leading to severe COVID-19 disease remain unclear, it is likely that an excessive innate immune response worsens it [164].

Emerging evidence suggests that SARS- CoV-2 highjacks mitochondria of immune cells, replicating within mitochondrial structures, and impairs mitochondrial dynamics, leading to cell death [165]. SARS-CoV-2 may manipulate mitochondrial function both directly and indirectly. SARS-CoV-2 enters the host cell by attaching to angiotensin-converting enzyme 2 (ACE2), followed by internalization and depletion of ACE2 receptors [166]. In the vascular endothelium, ACE2 converts angiotensin II to angiotensin. Thus, ACE2 depletion results in increased levels of angiotensin II, a pro-thrombotic, vaso-constrictive and pro-inflammatory peptide hormone that boosts cytoplasmic and mitochondrial ROS levels, causing oxidative stress and mitochondrial dysfunction (reviewed in [167]). This indirect action on mitochondria would promote the acute lung injury observed in severe cases of COVID-19 infection.

Besides this indirect effect, SARS-CoV-2 may directly manipulate the mitochondrial function through its accessory protein, Orf9b, which suppresses interferon I (IFN-I) responses by binding to TOM70, an outer membrane mitochondrial protein [168]. Notably, activation of IFN-I is a crucial event of the immune defense response against viral infection [169], thus its suppression would promote virus replication and COVID-19 disease. In addition, a recent study suggests that SARS-CoV-2 suppresses mitophagy [170], thus promoting accumulation of damaged mitochondria that may favor inflammation and cell death [167]. Viral ORFs, acting on host mitochondria, can cause release of mtDNA in the cytoplasm, thus activating mtDNA-induced inflammasome and suppressing innate and adaptive immunity [167]. Future investigations aimed at uncovering how viruses communicate with host mitochondria may provide a better understanding of COVID-19 pathology as well as alternative therapeutic strategies.

The role of mitochondria during the hyper-inflammatory state termed “cytokine storm” in COVID-19 disease remains to be clarified. A recent study proposed that the cytokine storm-mediated iron dysregulation, inducing ROS production and oxidative stress, and phenotypically appearing as hyperferritinemia, correlates with severity of the disease [171]. The authors also proposed that the inflammatory signals activate a vicious cycle that worsens mitochondrial oxidative damage and contributes to coagulopathy and ferroptosis [171]. Saleh et al. also suggested that COVID-19 infection not only causes intracellular mitochondria dysfunction, but also affects extracellular mitochondria, particularly platelets mitochondria, promoting blood clots and thrombosis. These extracellular mitochondria may serve as therapeutic targets to signal the severity of COVID-19 disease [171]. In addition to extracellular whole mitochondria, highly elevated levels of circulating mtDNA have been measured in the blood of COVID-19 patients with severe clinical manifestations of the disease [172], revealing emerging markers for COVID-19 therapy.

A schematic illustration of mitochondrial involvement in COVID-19 pathogenesis is reported in Figure 4.

## 5. Future Perspectives and Therapeutic Potential of Intercellular Mitochondrial Transfer

The recent experimental evidence that mitochondria can crosstalk and move beyond cell boundaries, in a variety of pathophysiological contexts, challenges the paradigm of intracellular segregation of mitochondria and mtDNA inheritance, opening a new era that is leading to the discovery of a more interconnected, dynamic and plastic nature of mitochondrial biology.

Identification of molecular mechanisms and signaling pathways leading and regulating intercellular mitochondrial transfer will be useful to promote its potential therapeutic application. In this context, future therapeutic strategies should be developed to improve the cell-to-cell transfer of organelles when it is functionally useful, or prevent it when detrimental, such as during the spread of viral infection or in the containment of neoplastic cells and their chemoresistance.

A new mitochondria-based therapeutic approach based on direct or systemic supplementation of mitochondria from autologous sources has been developed [173,174]. Such therapy to replace dysfunctional mitochondria improved cellular bioenergetics and reduced the oxidative stress [175,176]. Delivery of healthy mitochondria from the same patient could lead to an even better outcome [177].

Mitochondrial transplantation strategies provided promising results for the therapy of cardiac ischemia, neurodegeneration and ischemia reperfusion of liver (reviewed in [178]), though the factors involved in intercellular mitochondrial transfer remain to be characterized.

When elaborating a mitochondrial transfer/transplant-based strategy, it must be considered that the mtDNA and mitochondrial components are highly immunogenic owing to their bacterial origin.

Besides this factor, taking into account the leading role of mitochondria in any pathophysiological context, it should be carefully assessed whether the transfer of mitochondria promotes health or disease in different clinical settings.

## 6. Concluding Remarks

We provide here an overview of the current knowledge of the intercellular mitochondrial trafficking, debating mode and mechanisms underlying the process and its involvement in different diseases. As highlighted in this review, the cell-to-cell transfer of mitochondria represents a widespread event occurring in a variety of pathophysiological settings, although the molecular mechanisms mediating intercellular mitochondrial transfer and the signaling regulating this process remain yet poorly understood.

Further studies are needed to identify trigger factors that drive mitochondrial transfer, the mechanisms of formation of the network of intercellular bridges and connections in different transfer models and their potential use as targets in different clinical settings.

## Figures and Tables

**Figure 1 ijms-22-08312-f001:**
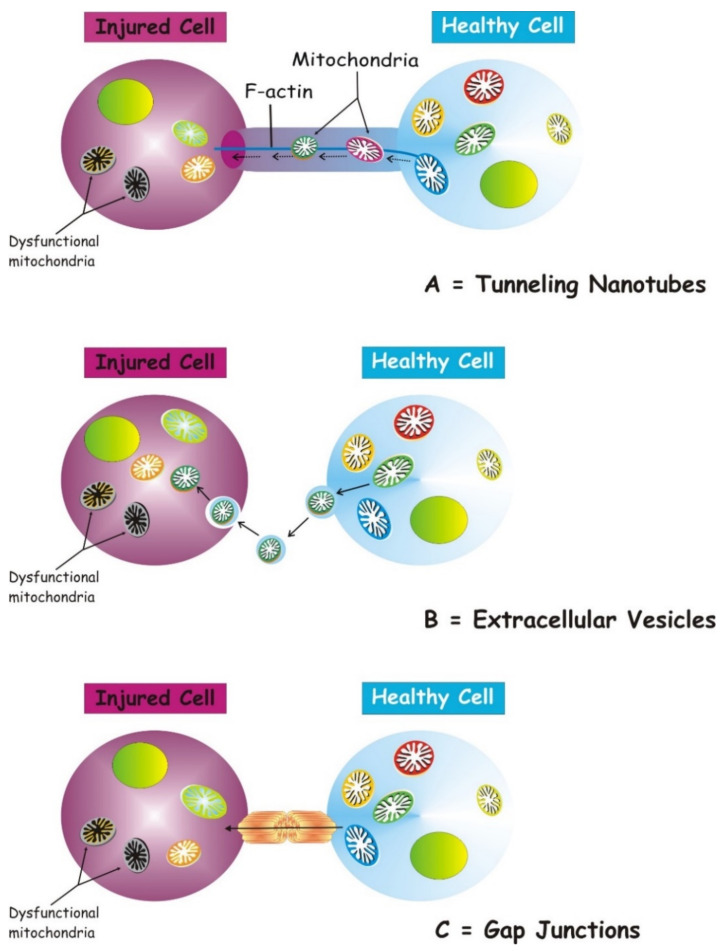
Three main forms of intercellular communication related to transcellular mitochondrial transfer. Under different injury signaling, three main forms of intercellular communication related to mitochondrial transcellular transfer can be formed: tunneling nanotubes (**A**), extracellular vesicles (**B**), and gap junctions. (**C**). For details see the text.

**Figure 2 ijms-22-08312-f002:**
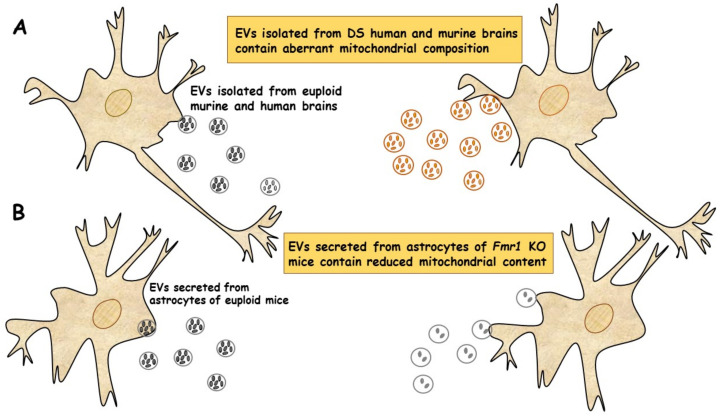
EVs of mitochondrial origin reflect mitochondrial dysfunction in DS and FXS. (**A**) Mitovesicles are a novel population of extracellular vesicles of mitochondrial origin altered in DS. Comparative analysis of EVs obtained from individuals with DS and Ts2 mouse model of DS showed higher numbers of mitovesicles with altered composition in the cerebral parenchyma, in both DS murine and human post-mortem brains [84]. (**B**) Depletion of mitochondrial components from extracellular vesicles secreted from astrocytes in a mouse model of FXS. Mitochondrial components contained in EVs derived from both cerebral cortices and astrocytes of *Fmr1* KO mice were found reduced [85].

**Figure 3 ijms-22-08312-f003:**
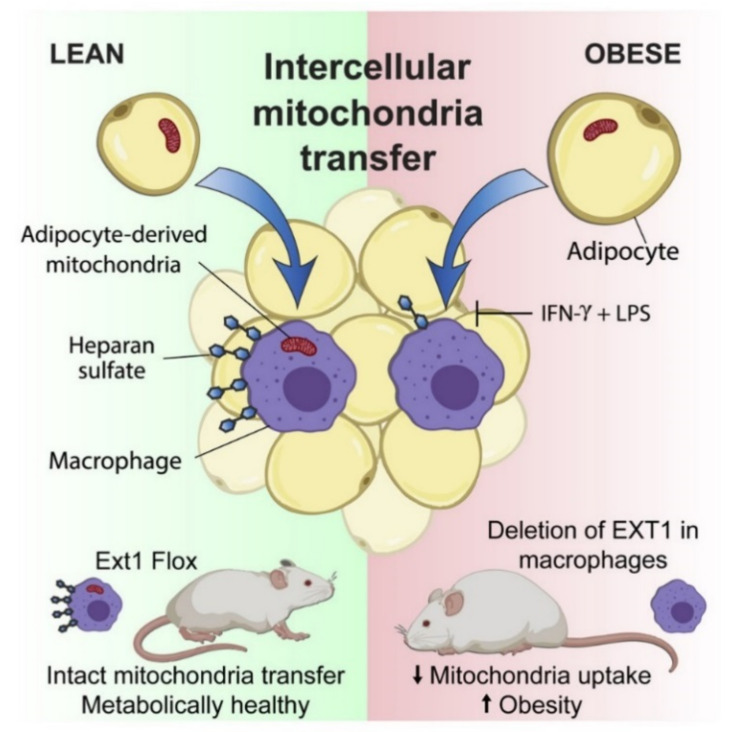
Intercellular mitochondrial transfer from adipocytes to macrophages. WAT-resident macrophages take up mitochondria from neighboring adipocytes in a heparan sulfate-dependent process that is impaired in obesity. Genetic disruption of mitochondria uptake by macrophages reduces energy expenditure and aggravates diet-induced obesity in mice, indicating that intercellular mitochondria transfer to macrophages promotes systemic metabolic homeostasis. Illustration from [128].

**Figure 4 ijms-22-08312-f004:**
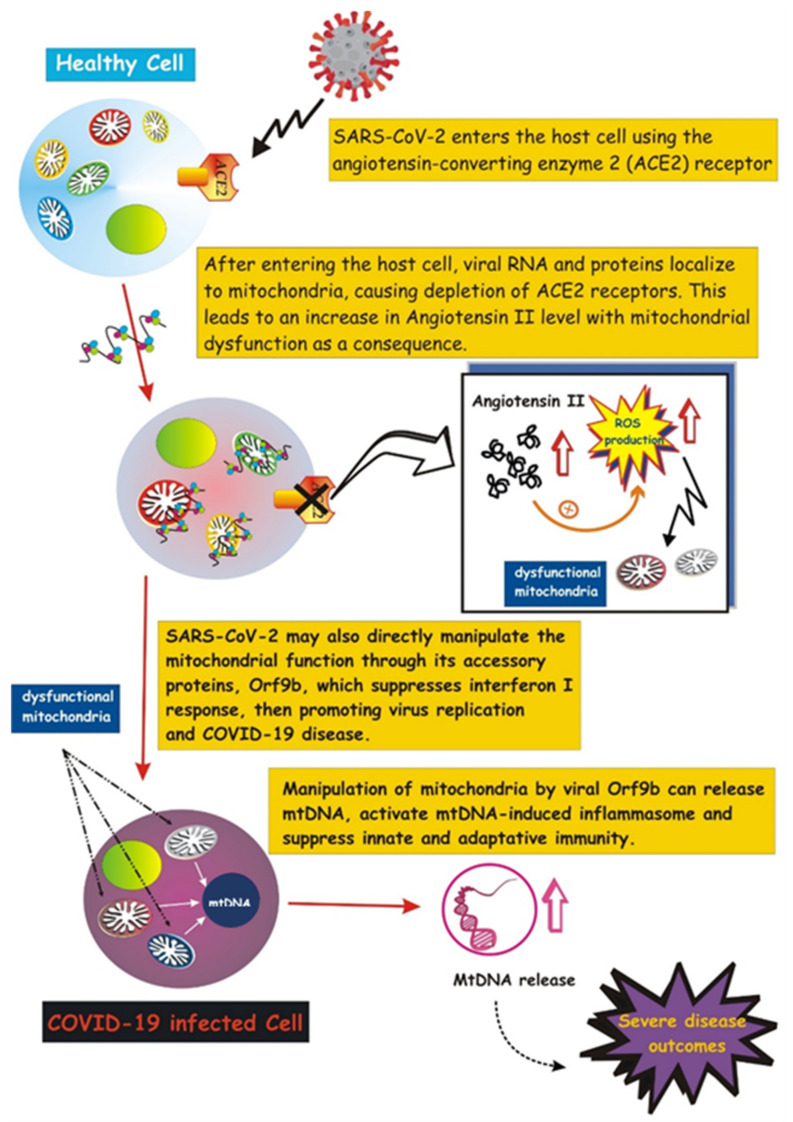
Involvement of mitochondria in COVID-19 pathogenesis. Schematic illustration showing the SARS-CoV-2 entry into the host cell by attaching to angiotensin-converting enzyme carboxypeptidase 2 (ACE2). Once inside the cell, viral RNA and proteins localize on mitochondria, resulting in ACE2 depletion and increase in Angiotensin-II, which, in turn, mediates mitochondrial dysfunction (more details in the text). SARS-2-CoV-2 targets the mitochondrial machinery also directly through one of its encoded proteins, Orf9, which promotes release of mtDNA and further mitochondrial dysfunction, leading to enhanced inflammasome activation.

## Data Availability

Not applicable.

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
