# Peer review of "Mitochondria Can Cross Cell Boundaries: An Overview of the Biological Relevance, Pathophysiological Implications and Therapeutic Perspectives of Intercellular Mitochondrial Transfer"

_ijms, 2021, doi:10.3390/ijms22158312_

Round 1

Reviewer 1 Report

In the present review the authors well hightlighted that the molecular mechanisms by which cells containing dysfunctional mitochondria acquire new mitochondria from other cells and, thus, the signaling regulating this process remain yet poorly understood.

The review is well organized and easy to read.

Some minor concerns:

  • The introduction should be more focused on the diseases mentioned in the review;
  • In Section 2. Modes and Cellular Structures mediating Intercellular Mitochondrial Transfer, the authors should add some references in this section;
  • Figure 1. Ingiury signals design is not appropriate in Panel A  and the direction of the trasfert needs to be insert correclty in A as already done in Panel B and C;
  • Provide an explanation of OXPHOS,  the first time is mentioned in the text;
  • The paragraph regarding Figure 3 is too vague concerning the EXT1 deletion; please explain in details;
  • In my opinion, Figure 3 is not necessary;
  • Section 4.1. Extracellular Circulating Cell-Free Mitochondria in Blood is a little bit redundant;
  • At the end of page 15, the sentence "SARS-CoV-2 may manipulate mitochondrial function indirectly, first by angiotensin- converting enzyme carboxy-peptidase 2 (ACE2) regulation of mitochondrial function, and once it enters the host cell, open-reading frames (ORFs) can directly manipulate mitochondrial function to evade host cell immunity and facilitate virus replication and COVID-19 disease" needs punctuation;
  • The sentence "Manipulation of host mitochondria by viral ORFs can release mtDNA in the cytoplasm, activate mtDNA-induced inflammasome and suppress innate and adaptive immunity. Under-standing the mechanisms underlying virus communication with host mitochondria may provide critical insights into COVID-19 pathology" needs references.
  • The sentence "Growing evidence correlate the accelerated progression of the disease in COVID-19 patients to the hyper-inflammatory state termed as the “cytokine storm” involving major systemic perturbations. These include iron dysregulation manifested as hyperferritinemia associated with disease severity. Iron dysregulation induces ROS production and pro-motes oxidative stress [166]" has been placed in the text not correctly. My suggestion is to postpone the sentence after "Many questions remain unanswered about the role of the mitochondria during the inflammatory “cytokine storm” in COVID-19 patients" and not to separate the sentences in a new paragraph.
  • Figure 4 is not appropriate to the sentence "SARS-CoV-2 may manipu-late mitochondrial function indirectly, first by angiotensin- converting enzyme carboxy-peptidase 2 (ACE2) regulation of mitochondrial function, and once it enters the host cell, open-reading frames (ORFs) can directly manipulate mitochondrial function to evade host cell immunity and facilitate virus replication and COVID-19 disease".
  • I agree with the authors of the relevance  of the topic of their review with the current state of  Sars-Cov-2 virus pandemic situation, but other medically important bacteria and viruses, such as the influenza virus, human immunodeficiency virus (HIV) and herpes simplex virus, can evade host immunity and avoid pharmaceutical targeting by using TNTs to pass their genomes to naive cells. Please, mention TNTs in other bacterial/viral infection;
  • Some English oversights should be revised

Author Response

In the present review the authors well highlighted that the molecular mechanisms by which cells containing dysfunctional mitochondria acquire new mitochondria from other cells and, thus, the signalling regulating this process remain yet poorly understood.

The review is well organized and easy to read.

Some minor concerns:

  • The introduction should be more focused on the diseases mentioned in the review;

We thank the reviewer for her/his suggestion, however given the heterogeneity of the different pathophysiological contexts reported in the review, the authors have preferred to introduce and describe the diseases mentioned in the following Section 3, each in their own dedicated paragraph, rather than focus on them in the Introduction. In the Introduction we have only modified the last sentence: " In this review, we will provide an overview of the up-to-date knowledge regarding the intercellular transfer of mitochondria, focusing on the cellular and molecular mechanisms mediating this process, discussing its biological relevance and involvement in different pathophysiological contexts,  -ranging from immuno-metabolic diseases, neurogenerative/neurodevelopmental disorders, and viral infections to cancer-  and highlighting  the potential therapeutic targeting of horizontal cell-to-cell transfer of mitochondria for the treatment of diseases characterized by mitochondrial dysfunction.”

  • In Section 2. Modes and Cellular Structures mediating Intercellular Mitochondrial Transfer, the authors should add some references in this section;

New references have been added in this section

  • Figure 1. Injury signals design is not appropriate in Panel A and the direction of the transfer needs to be insert correctly in A as already done in Panel B and C;

Figure 1 has been changed according to the reviewer’s suggestion, deleting “the injury signals design” in Panel A and inserting the arrows inside the TNT showing the direction of the mitochondrial transfer.

  • Provide an explanation of OXPHOS, the first time is mentioned in the text;

Explanation of OXPHOS abbreviation has been provided the first time it was mentioned

  • The paragraph regarding Figure 3 is too vague concerning the EXT1 deletion; please explain in details; In my opinion, Figure 3 is not necessary;

The function of EXT1 (exostosin 1) gene encoding a copolymerase involved in heparan sulfate biosynthesis and the effect of its deletion on lipid homeostasis have been better explained.

Regarding the comment on Figure 3, the authors would like to keep it in the review, considering very explicative this picture showing the graphical abstract of the very recent in vivo study of Brestoff et al. (2021) since it provides evidence for the first time of the existence of intercellular mitochondria transfer from adipocytes to macrophages, regulating white adipose tissue homeostasis and resulted impaired in obesity. We received license from Elsevier for the publication of this copyrighted illustration (License Number: 5094081011375)

  • Section 4.1. Extracellular Circulating Cell-Free Mitochondria in Blood is a little bit redundant;

Section 4.1. has been shortened

  • At the end of page 15, the sentence "SARS-CoV-2 may manipulate mitochondrial function indirectly, first by angiotensin- converting enzyme carboxy-peptidase 2 (ACE2) regulation of mitochondrial function, and once it enters the host cell, open-reading frames (ORFs) can directly manipulate mitochondrial function to evade host cell immunity and facilitate virus replication and COVID-19 disease" needs punctuation;

The sentence has been changed, describing in more detail the indirect and direct manipulations of the mitochondria by SARS-CoV-2.

  • The sentence "Manipulation of host mitochondria by viral ORFs can release mtDNA in the cytoplasm, activate mtDNA-induced inflammasome and suppress innate and adaptive immunity. Under-standing the mechanisms underlying virus communication with host mitochondria may provide critical insights into COVID-19 pathology" needs references.

References have been added after the sentence

  • The sentence "Growing evidence correlate the accelerated progression of the disease in COVID-19 patients to the hyper-inflammatory state termed as the “cytokine storm” involving major systemic perturbations. These include iron dysregulation manifested as hyperferritinemia associated with disease severity. Iron dysregulation induces ROS production and pro-motes oxidative stress [166]" has been placed in the text not correctly. My suggestion is to postpone the sentence after "Many questions remain unanswered about the role of the mitochondria during the inflammatory “cytokine storm” in COVID-19 patients" and not to separate the sentences in a new paragraph.

The text has been changed according to the reviewer’s suggestion

  • Figure 4 is not appropriate to the sentence "SARS-CoV-2 may manipulate mitochondrial function indirectly, first by angiotensin- converting enzyme carboxy-peptidase 2 (ACE2) regulation of mitochondrial function, and once it enters the host cell, open-reading frames (ORFs) can directly manipulate mitochondrial function to evade host cell immunity and facilitate virus replication and COVID-19 disease".

Figure 4 has been modified reporting in more detail both indirect and direct SARS-CoV-2 manipulation of mitochondrial function. The text has also been modified.

  • I agree with the authors of the relevance of the topic of their review with the current state of Sars-Cov-2 virus pandemic situation, but other medically important bacteria and viruses, such as the influenza virus, human immunodeficiency virus (HIV) and herpes simplex virus, can evade host immunity and avoid pharmaceutical targeting by using TNTs to pass their genomes to naive cells. Please, mention TNTs in other bacterial/viral infection;

Mention and some examples of other viral infections manipulating mitochondrial function and using TNTs to transfer their genomes to naive cells has been added in paragraph 4.3., in agreement with the reviewer's suggestion.

  • Some English oversights should be revised

A careful check of English style has been made

Reviewer 2 Report

A review by Valenti et al. discusses the intercellular trafficking of mitochondria, mode and mechanisms underlying the process and its involvement in different diseases.

Comments:

  1. Authors often use “Growing evidences” in the text of the manuscript. However, the authors either use only one reference, or do not cite references to literary sources at all. This does not look correct and rather speculative. The growing body of evidence has to be backed up by several literary references.
  2. The need for Figure 2 is not clear. This figure does not provide additional information. From the figure, we see that vesicles with green mitochondria release from green cells, and vesicles with red mitochondria release from red cells. Why should the reader have to guess what this color differentiation means and how green mitochondria differ from red ones?
  3. Authors should check the citation of sources. For example, the authors write: “Larger EVs such as MVs can even contain entire mitochondria [38]”. However, this source does not mention this fact. Another example: “Cancer cells can release entire mitochondria or their components, such as mtDNA, ATP, cytochrome c, or formylated peptides, to the extracellular space [96]” Reference 96 does not mention this either. Moreover, I would like to wish the authors to refer to the original articles, and not to review.
  4. It is necessary to write the Conclusion section.
  5. The word of “process” in the sentence should be corrected. “…Specifically in neurons, mitochondria are transported anterograde from the cell body to the processes of the cell, or retrograde back to cell body…” The check of English style is also required.

Author Response

A review by Valenti et al. discusses the intercellular trafficking of mitochondria, mode and mechanisms underlying the process and its involvement in different diseases.

Comments:

  1. Authors often use “Growing evidences” in the text of the manuscript. However, the authors either use only one reference, or do not cite references to literary sources at all. This does not look correct and rather speculative. The growing body of evidence has to be backed up by several literary references.

We agree with the reviewer for his/her observation and we have changed the adjective, where necessary

  1. The need for Figure 2 is not clear. This figure does not provide additional information. From the figure, we see that vesicles with green mitochondria release from green cells, and vesicles with red mitochondria release from red cells. Why should the reader have to guess what this colour differentiation means and how green mitochondria differ from red ones?

The authors have modified Figure 2, by eliminating the colour (green/red) differentiation - resulted unclear to the reviewer - which had the visual purpose to distinguish in green the representation of neurons and astrocytes in wt mice, from nervous cells in mouse models of DS and Fragile-X respectively, represented in red). To highlight the differences, the figure has been now modified showing how the EVs isolated in the mouse model of DS are more numerous but with an aberrant mitochondrial content compared to wt mice, as well as the EVs secreted by astrocytes of KO-Fmr1 mice show a lower content of mitochondria than wt. 

  1. Authors should check the citation of sources. For example, the authors write: “Larger EVs such as MVs can even contain entire mitochondria [38]”. However, this source does not mention this fact. Another example: “Cancer cells can release entire mitochondria or their components, such as mtDNA, ATP, cytochrome c, or formylated peptides, to the extracellular space [96]” Reference 96 does not mention this either. Moreover, I would like to wish the authors to refer to the original articles, and not to review.

A careful check of sources of citated references has been made

  1. It is necessary to write the Conclusion section.

A final section of Concluding remarks has been added

  1. The word of “process” in the sentence should be corrected. “…Specifically in neurons, mitochondria are transported anterograde from the cell body to the processes of the cell, or retrograde back to cell body…” The check of English style is also required.

In the sentence reported above the word “processes” has been changed with “axonal extensions”; a careful check of English style has been made